# Effect of Spherical WC Content on the Microstructure and Properties of SiCp Aluminium Composite Material

Xinyi Feng, Xiao Li * , Fei Wang, Zengzhi Liu and Wenping Wang

School of Mechanical Engineering, Xinjiang University, Urumqi 830047, China; 107552101393@stu.xju.edu.cn (X.F.); 107552001095@stu.xju.edu.cn (F.W.); 107552101392@stu.xju.edu.cn (Z.L.); 107552204287@stu.xju.edu.cn (W.W.)
* Correspondence: likebaijia@126.com

**Abstract:** In this paper, SiCp aluminium matrix composites were used as the matrix, and AlSi10Mg powder, which has a relatively similar coefficient of thermal expansion to that of the matrix, was used to prepare laser cladding Al-based coatings. The results show that the optimal process parameters are $P$ = 4400 W, $V_f$ = 11.3 g·min$^{-1}$, and $V_S$ = 1800 mm·min$^{-1}$, and, although the hardness of the coatings is lower than the hardness of the substrate, it reduces the generation of defects such as cracks and porosity. With the increase in WC reinforced phase and the hardness of the coatings, wear resistance increases, the granular cytocrysts are transformed into rod-like cytocrysts, and at the same time generate the dendritic crystals, and the undergo grain refining and generate the new phases such as Al$_4$C$_3$, Al$_4$SiC$_4$. There is no obvious defect in AlSi10Mg + 40%WC coatings, the macro morphology of the coatings is good, there is no spalling in the friction wear morphology, and the wear resistance is excellent, but there are obvious cracks and obvious spalling in the coatings of AlSi10Mg + 60%WC. Compared to the matrix hardness of 171.61 HV, the hardness of the 20%WC cladding layer increased by a factor of 1.06, while the hardness of the 40%WC cladding layer increased by a factor of 1.65 and that of the 60%WC cladding layer increased by a factor of 1.8. In terms of wear, compared to a substrate wear amount of 9.36 mg, the wear for the 20%WC cladding layer was reduced to 6.13 mg (34.5% less than the substrate), for the 40%WC cladding layer it was reduced to 4.58 mg (51.06% less than the substrate), and for the 60%WC cladding layer it was reduced to 7.35 mg (21.47% less than the substrate). The quality of the coatings decreases although the hardness is higher than that of AlSi10Mg + 40%WC. The comprehensive performance of AlSi10Mg + 40%WC coatings is optimal.

**Keywords:** laser cladding; pulsed electric field; preheating; wear performance

## 1. Introduction

Aluminium content is the most abundant metallic substance in the world, and aluminium alloys have become the most widely used metallic materials worldwide, with their excellent fracture toughness, high specific strength, and good forming and metallurgical properties attracting the attention of industries in various countries. However, its inherent drawbacks, such as low hardness, low wear resistance, and low modulus of elasticity, make aluminium alloys increasingly unable to meet the needs of emerging fields of industry [1,2]. For this reason, researchers and scholars have thought of adding certain substances that can enhance the properties of traditional aluminium alloys to prepare composites with excellent properties, in which various reinforcing particles, including SiC particles, have the advantages of low density (3.21 g/cm$^3$), high modulus of elasticity and hardness, good chemical stability, wear resistance, and low cost of preparation, and at the same time, have good chemical compatibility with aluminium alloys. The comprehensive performance of SiC particles makes them the most commonly used in aluminium matrix composites [3,4]. SiC particles reinforced aluminium matrix composites have stronger properties of elastic modulus, wear resistance, specific strength, and dimensional stability compared to traditional aluminium alloys, and the addition of different volume fractions of SiC particles

has different effects on their properties, which are widely used in the fields of electronic packaging and aerospace and automotive manufacturing.

AlSi10Mg [5–8] as an aluminium–silicon alloy is used in the field of material forming because of its good castability and corrosion resistance. In recent years, researchers in the field of civil aviation widely used SLM technology [9–11] for its complex shape, and relevant research shows that the mechanical properties of SLM-forming alloys [12–15] are better than those of the traditional casting products [16]. At the same time, in terms of wear resistance, the most commonly used surface texture technology is laser surface texture, which can greatly improve friction performance [17,18]. However, many factors limit the development of laser additive manufacturing; AlSi10Mg alloys, such as aluminium alloys are prone to oxidation at high temperatures [19], high thermal conductivity, and laser reflectivity [20], and defects are prone to porosity, cracks, and other defects. How to effectively reduce coating cracks is a hot topic in the field of laser cladding, Li [21] In the laser cladding process, pulse current is applied to effectively mitigate coating cracks and enhance coating hardness and wear resistance. At the same time, because of the low heat input of laser cladding and a high degree of automation, it is suitable for metal repair technology. Al-based alloy cladding materials are mainly used in the laser cladding repair of aluminium alloy products, in order to adapt to the economy of saving, research on the laser cladding repair technology of aluminium alloy is particularly necessary. Scholars have studied the laser repair technology; Meinert and Bergan [22] first reported on the laser cladding repair of aluminium alloys, in 6061 and 7075 aluminium alloy surface cladding 4000 series aluminium alloys, the tensile strength of the repaired specimens for the original 6061-T6 and 7075-T651 specimens were 61% and 64%, respectively. Corbin et al. [23] carried out laser cladding of 6061-T6-aluminium alloy powder on the surface of 6061 aluminium alloy powder laser cladding on the surface of 6061T6-aluminium alloy; they found that the hardness of the coating is lower than the substrate, and there are defects such as porosity, but the substrate and the coating powder properties are similar to reduce the generation of cracks, but did not continue to carry out the coating performance enhancement of the relevant research.

Wear calculations are very important in order to predict the life cycle of a certain component, and thus schedule the forehand replacement or prevent the failure [24]. WC [25–27] is a kind of ceramic particle, with high hardness and wear resistance, and has good electrical and thermal conductivity, WC has good chemical properties [28,29], and its powder is often used for cemented carbide reinforcing materials. With the development of laser additive manufacturing, more and more scholars will also be WC ceramic particles added to its coating powder as a reinforcing phase, C element, and W element in the laser high temperature will occur with other elements [30,31]. C and W elements will react with other elements under the high temperature of the laser to strengthen the coating with a solid solution, and then improve the hardness, wear resistance, and corrosion performance of the coating surface [32–34]. In Yu [35], AlSi5Cu1Mg aluminium alloy surface laser cladding Ni60 + WC powder, the cladding layer microhardness of the substrate is about 9~11 times. Qing et al. [36] studied the microstructure, microhardness, and wear resistance of 60% WC-reinforced FeCoNiCr composite coatings prepared by laser cladding. The study showed that the composite coatings have good forming quality, better microhardness, and wear resistance.

Although the desired properties can be achieved through laser cladding of metal powders with varying characteristics on the substrate, the coating may develop cracks due to its high coefficient of thermal expansion. This issue significantly impacts the binding properties of the coating itself. In this study, we aim to address these concerns by following previous scholars' theories on repairing cracks and defects in original aluminum alloy substrates through laser cladding different types of Al-based powders onto an Al substrate. By selecting AlSi10Mg powder with a similar thermal expansion coefficient as the substrate and incorporating the WC reinforcing phase to enhance surface wear resistance and hardness properties, we can effectively minimise crack formation.

This approach has potential cost-saving implications for commercial applications while expanding the utilisation of aluminum alloys. The key distinction between our research and others lies in our focus on improving aluminum alloy laser cladding properties to reduce crack occurrence, thereby providing a valuable reference for preparation processes and theoretical foundations.

## 2. Experimental

### 2.1. Experimental Materials

Experiments selected by the AVIC Xi'an Aircraft Manufacturing Company Limited Materials Research Institute, Xi'an, China provided by the 15% SiCp/2009Al composition (see Table 1) and size 150 mm × 60 mm× 10 mm, before using, 400 #, 800 #, and 1000 # of sandpaper will be used to sand the surface to a bright, remove the surface of the oil and oxidation, and then put into the ultrasonic vibration instrument filled with anhydrous ethanol. Cleaned 30 min, and then placed in a drying box for 120 min standby, the powder material selection of 100–250 mesh AlSi10Mg; for powder composition see Table 2. Then, mix 20%, 40%, and 60% WC spherical ceramic powder; powder composition is shown in Table 3. The mixing process is conducted using a KQM-Z/B-type planetary ball mill for a duration of 2 h to ensure thorough and uniform blending. The ratio of balls to material is maintained at 3:1, while the milling speed is set at 500 r/min. Subsequently, the mixed powder is transferred into a drying oven for a period of 60 min in order to eliminate any moisture content present. Once dried, it is then placed onto a tray and allowed to cool down naturally until reaching room temperature before being carefully stored in an air-tight bag.The SEM images of AlSi10Mg powder and WC powder are depicted in Figure 1.

**Table 1.** Chemical composition of 15% SiCp/2009Al substrate (wt.%).

| Cu | Mg | Fe | Si | Al |
|---|---|---|---|---|
| 4.2 | 1.6 | 0.05 | 0.25 | Bal. |

**Table 2.** Chemical composition of AlSi10Mg coated powder (wt.%).

| Si | Fe | Cu | Mg | Mn | Ni | Al |
|---|---|---|---|---|---|---|
| 9.880 | 0.051 | 0.005 | 0.400 | 0.009 | 0.180 | Bal. |

**Table 3.** Chemical composition of WC ceramic reinforced phase powders (wt.%).

| Fe | Si | Mg | Al | k | Na | Ca | S | Mo | W | C |
|---|---|---|---|---|---|---|---|---|---|---|
| 0.02 | 0.003 | 0.002 | 0.002 | 0.0015 | 0.0015 | 0.002 | 0.002 | 0.01 | Bal. | Bal. |

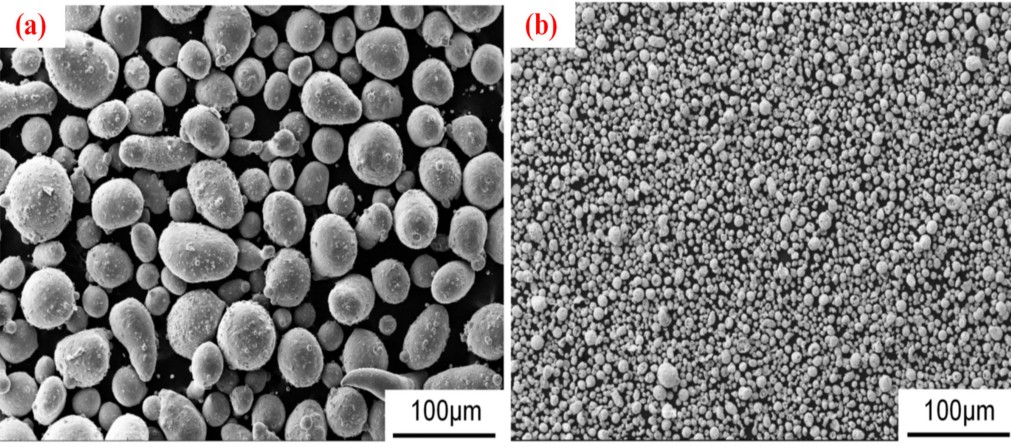

**Figure 1.** (**a**) SEM (Scanning Electron Microscope) morphology of spherical AlSi10Mg. (**b**) SEM (Scanning Electron Microscope) morphology of spherical WC.

## 2.2. Experimental Methods

The cladding layer was prepared by an 8 kW JPT-6000 IPG continuous wave fiber laser from Shaanxi Yunding Optoelectronic Technology Co, Ltd., Xi'an, China with a 1080 nm wavelength. The spot diameter is 2 mm and was used to process the AlSi10Mg coated single-pass specimen; with coaxial powder feeding processing, the protective gas was argon, and the gas flow rate was set at 13 L/min. The laser schematic is shown in Figure 2.

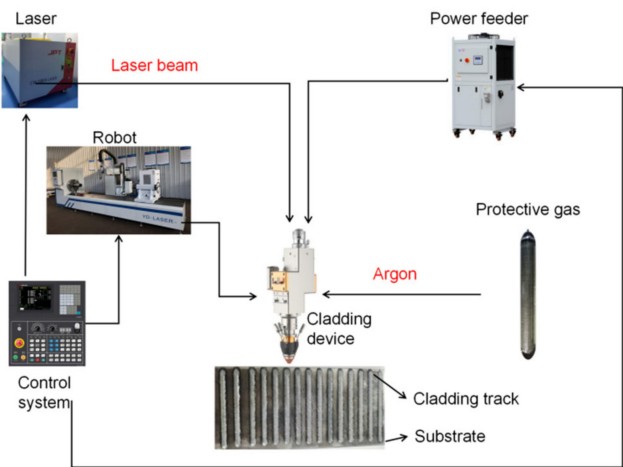

**Figure 2.** Component modules of the fusion cladding system.

Firstly, a one-factor orthogonal test of AlSi10Mg coating was carried out, and the effects of different laser power, powder feeding rate, and scanning rate on the coating quality could be analysed through the one-factor experiment, and the orthogonal test method is an experimental method to study the combination of multi-factors and multi-levels, whose main function is that it can be substituted for the full-scale test by a small number of tests, which saves time and cost. We selected the better process parameters as the intermediate level value and establish three factors and three levels of orthogonal test; the value of each factor is shown in Table 4.

**Table 4.** Orthogonal test factor level table.

| Factors | (A) Laser Power/(w) | (B) Powder Feeding Rate/(g·min$^{-1}$) | (C) Scanning Speed/(mm·min$^{-1}$) |
| --- | --- | --- | --- |
| Level 1 | 4200 | 6.78 | 1800 |
| Level 2 | 4400 | 9.04 | 2000 |
| Level 3 | 4600 | 11.3 | 2200 |

According to the number of levels of each factor to choose a single level of orthogonal table L$_9$ (3*3), the level of each factor based on the table arrangement to obtain the orthogonal test parameter table, each line of the factors to take the value of the combination of a group of test parameters [37], for a total of 9 groups of tests, as shown in Table 5.

Figure 3 shows the macroscopic morphology of the single-pass coating cross-sections obtained under the orthogonal test parameters described above, and it can be seen that the cross-sectional morphology of the different groups varies relatively widely, and the dilution rates of all specimens are generally high. Different scanning rates change the laser scanning dwell time, which also changes the amount of powder obtained from the substrate in disguise. As can be seen from Figure 3, under the variation of the current parameter gradient, when the power is certain, the scanning rate has a greater influence on the melt height and depth than the scanning speed, and the difference in this influence is gradually obvious when accompanied by the power increase. This is because the change of 200 mm·min$^{-1}$ powder feeding rate on the coating of the same position of the scanning rate of the substrate to obtain a greater amount of powder; when the power gradually

increases, the heat in the melt pool rises, the unmelted powder decreases, and the powder utilisation increases; compared to the lower power, the increase in the amount of powder permitted by the higher power will also be more, thus producing a significant difference. When $P$ = 4200 W and 4600 W, the coatings are produced with more porosity as seen in Figure 3 1–3,7,9, while when $P$ = 4400 W, Figure 3 5,6 have no obvious porosity defects, indicating that the appropriate power parameters have a greater impact on the generation of defects.

**Table 5.** Parameter list of orthogonal test.

| No. | A | B | C |
|---|---|---|---|
| 1 | 1 (4200) | 1 (6.78) | 1 (1800) |
| 2 | 1 | 2 (9.04) | 2 (2000) |
| 3 | 1 | 3 (11.3) | 3 (2200) |
| 4 | 2 (4400) | 1 | 2 |
| 5 | 2 | 2 | 3 |
| 6 | 2 | 3 | 1 |
| 7 | 3 (4600) | 1 | 3 |
| 8 | 3 | 2 | 1 |
| 9 | 3 | 3 | 2 |

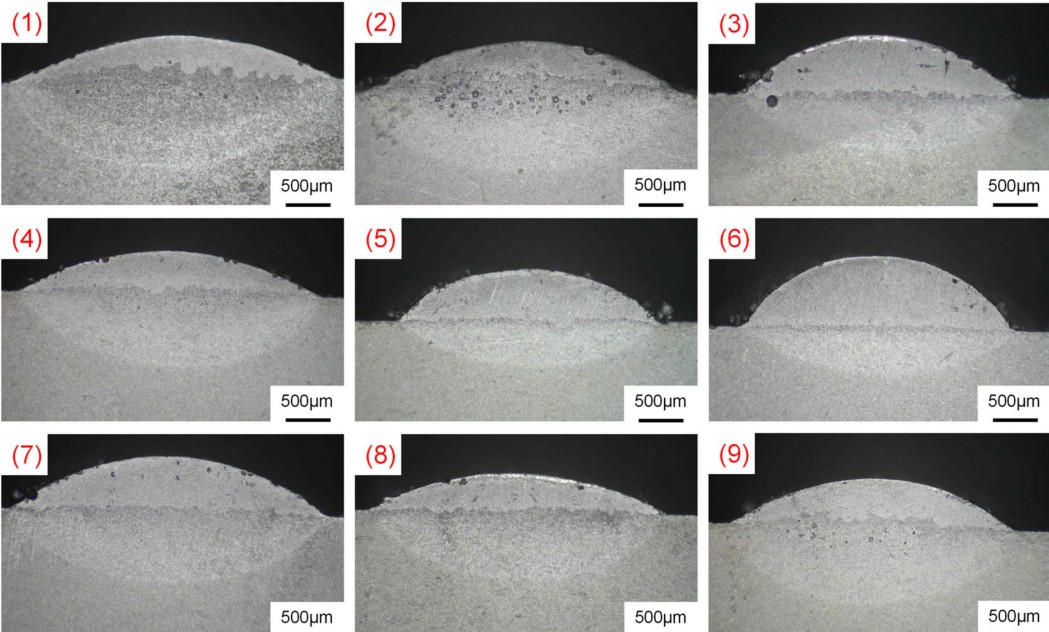

**Figure 3.** Macroscopic morphology of single-pass coating cross-section with orthogonal test parameters. (1) $P$ = 4200 W, $V_f$ = 6.78 g·min$^{-1}$, $V_S$ = 1800 mm·min$^{-1}$. (2) $P$ = 4200 W, $V_f$ = 9.04 g·min$^{-1}$, $V_S$ = 2000 mm·min$^{-1}$. (3) $P$ = 4200 W, $V_f$ = 11.3 g·min$^{-1}$, $V_S$ = 2200 mm·min$^{-1}$. (4) $P$ = 4400 W, $V_f$ = 6.78 g·min$^{-1}$, $V_S$ = 2000 mm·min$^{-1}$. (5) $P$ = 4400 W, $V_f$ = 9.04 g·min$^{-1}$, $V_S$ = 2200 mm·min$^{-1}$. (6) $P$ = 4400 W, $V_f$ = 11.3 g·min$^{-1}$, $V_S$ = 1800 mm·min$^{-1}$. (7) $P$ = 4600 W, $V_f$ = 6.78 g·min$^{-1}$, $V_S$ = 2200 mm·min$^{-1}$. (8) $P$ = 4600 W, $V_f$ = 9.04 g·min$^{-1}$, $V_S$ = 1800 mm·min$^{-1}$. (9) $P$ = 4600W, $V_f$ = 11.3 g·min$^{-1}$, $V_S$ = 2000 mm·min$^{-1}$.

The melt width and dilution rate of each group of tests are shown in Table 6. The variation rule of melt width and dilution rate with the level of individual factors is not clear, and in order to further explore the influence of each factor on the melt width and dilution rate, the extreme difference analysis is used to further illustrate.

The extreme difference analysis table shown in Table 7, $R_j$ table on the extreme difference of factor j, reflecting the magnitude of changes in the test indicators with the level of the factor, is a measure of the degree of influence of the factor. According to the size of

the extreme difference $R_j$, you can judge the influence of the factors of the primary and secondary. The formula for calculating the extreme difference $R_j$ is shown in Equation (1), where $K_{jm}$ represents the factor j to take the level m, the sum of the test indicators.

$$R_j = \max\left(\overline{K_{J1}}, \overline{K_{J2}}, \ldots \overline{K_{Jm}}\right) - \min\left(\overline{K_{J1}}, \overline{K_{J2}}, \ldots \overline{K_{Jm}}\right) \tag{1}$$

**Table 6.** Melt width and dilution rate of orthogonal test coatings.

| No. | Dilution Rate/% | Melt Wide |
|-----|-----------------|-----------|
| 1 | 68.2 | 5.04 |
| 2 | 59.5 | 4.97 |
| 3 | 49.7 | 4.37 |
| 4 | 63.5 | 4.71 |
| 5 | 46.3 | 4.04 |
| 6 | 38.3 | 4.18 |
| 7 | 57.0 | 4.96 |
| 8 | 65.0 | 4.55 |
| 9 | 64.6 | 5.41 |

**Table 7.** Orthogonal test extreme variance analysis table.

| | | A | B | C |
|---|---|-----|-----|-----|
| | K1 | 14.38 | 14.71 | 13.77 |
| | K2 | 12.93 | 13.56 | 15.09 |
| | K3 | 14.92 | 13.96 | 13.37 |
| | $\overline{K_1}$ | 4.79 | 4.90 | 4.59 |
| Melt wide | $\overline{K_2}$ | 4.31 | 4.52 | 5.03 |
| | $\overline{K_3}$ | 4.97 | 4.65 | 4.46 |
| | *Kmax* | 4.97 | 4.90 | 5.03 |
| | *Kmin* | 4.31 | 4.52 | 4.46 |
| | R | 0.66 | 0.38 | 0.57 |
| | K1 | 177.40 | 188.70 | 174.40 |
| | K2 | 151.00 | 170.80 | 187.60 |
| | K3 | 186.60 | 155.50 | 153.00 |
| | $\overline{K_1}$ | 59.13 | 62.90 | 58.13 |
| Dilution rate/% | $\overline{K_2}$ | 50.33 | 56.93 | 62.53 |
| | $\overline{K_3}$ | 62.20 | 51.83 | 51.00 |
| | *Kmax* | 62.20 | 62.90 | 62.53 |
| | *Kmin* | 50.33 | 51.83 | 51.00 |
| | *R* | 11.87 | 11.07 | 11.53 |

As can be seen from Table 8, for the coating melt width RA > RC > RB and the dilution rate RA > RC > RB, with reference to the method of polar deviation can be seen, the different process parameters for the coating melt width of the influence of the size of laser power is the largest, followed by the scanning rate, the smallest rate of powder delivery; on the dilution rate of the influence the size of the laser power is the largest, the scanning rate of the second, the smallest rate of powder delivery, it can be seen clearly that, in the test, the laser power of the largest influence on the coating is the influence of the coating melt width and the dilution rate of the primary factors.

**Table 8.** Process parameters after orthogonal optimisation.

| Laser Power/(w) | Powder Feeding Rate/(g·min$^{-1}$) | Scanning Speed/(mm·min$^{-1}$) |
|-----------------|-----------------------------------|--------------------------------|
| 4400 | 11.3 | 1800 |

For the evaluation index of the coating, the moderate melt width and the lowest dilution rate are used as the selection principle of the parameters. For factor A, the influence of melt width and dilution rate of the coating is in the first place, the comprehensive

consideration of the coating morphology and dilution rate selection of $A_2$; for factor B, the melt width of the coating and the dilution of the rate of influence of the coating are in the last place, the comprehensive consideration of the selection of $B_3$; for factor C, the melt width of the coating and the dilution rate of the influence of the coating are in the middle place, the comprehensive consideration of the selection of $C_1$, and optimal combination of the analysis of the polarity of the difference for $A_2B_3C_1$.

After the end of laser cladding, the sample is cut into an $8 \times 8 \times 8$ mm size, polished with 800#, 1000#, and 1200# sandpaper, and then configured with aluminium-based electrolytic polishing solution (volume ratio of $HClO_4:CH_3CH_2OH = 1:9$) and electrolytic polishing with the electrolytic polishing machine model KXN-305D, set the voltage at 25 v, the electrolysis time is 20 s. During the electrolysis period, the electrolyte was stirred with a magnetic stirrer, and liquid nitrogen was added to cool down the temperature to prevent the electrolysis from increasing with time; the temperature increased, the resistance increased, and the specimen was blackened by over-electrolysis, which affected the amount of electrolytic quality.

An industrial microscope was used to analyse the macroscopic morphology and dilution rate of the coating cross-section; a Bruker D8 Advance X-ray diffractometer (XRD) was used to scan the coating for physical analysis; and JSM-7610F plus Scanning Electron Microscope (SEM) and energy spectrometry (EDS) were used to observe the microscopic morphology and elemental distribution. The microhardness was measured by Huayin HV-1000A digital microhardness tester with an applied force of 200 N and a dwell time of 15 s; the friction factor was tested by MFT-5000 friction and wear tester, and the dry friction experiment was carried out by using $Si_3N_4$ ceramic ball as the friction partner under a load of 60 N, with a loading time of 20 min, a frequency of 2 Hz, and a scratch distance of 10 mm.

## 3. Results and Discussion

### 3.1. Macroscopic Morphology Analysis

Through the above single-channel orthogonal experiment and range analysis, it is concluded that sample No. 6 ($A_2B_3C_1$) is the best process parameter, and the data are shown in Table 8.

Figure 4 shows the macroscopic morphology cross-section of AlSi10Mg coated powder and WC ceramic powder added with different proportions under an industrial microscope.

As can be seen from the figure, under the optimal orthogonal process parameters, the surface morphology of the AlSi10Mg coatings without added WC ceramic powder is good, the surface is flat and smooth, and there are no defects such as cracks and porosity, while the coatings with 60% WC addition has obvious cracks and porosity defects, and the moulding quality is poor. Is due to the thermal stress easily generated during the laser cladding process, and the difference between the thermal expansion coefficient of WC ceramic powder and the thermal expansion coefficient of the substrate, along with the continuous improvement of the WC content, the difference between the thermal expansion coefficient of WC ceramic powder and the substrate is also increasing. In the case of the same laser power, there is a large amount of unmelted powder in the 60% WC cladding group, and these particles are prone to produce a source of cracks, and, ultimately, cracks are generated. In addition, the changes in the composition of the alloy also caused a phase transition that increased the phase transition stress in the cladding material.

In the process of rapid cooling of laser cladding, the thermal stress of the cladding layer exceeds the yield strength of the material itself, and the gas that cannot be discharged by the rapid cooling of laser cladding will turn into porosity. So, the combined effect of the escape velocity of the gas and the speed of the laser condensation is one of the main reasons for the generation of porosity, and the WC makes the viscosity of the molten pool rise, which in turn restricts the escape of the gas, and a large number of WC particles into the molten pool. Burnout decomposition will occur to form $W_2C$ and C, and C atoms and O atoms, in the air reaction precipitation of CO and $CO_2$, the chemical formula seen in

Formulas (2) and (3) [36], making 60% of WC additions to the coatings produced defects such as pores and cracks, from the coatings colour, with the increase in WC powder content, the coatings metal gloss is reduced, which is due to the WC ceramic powder itself was black. This is due to the WC ceramic powder itself being black, with the increase in the content of WC powder, AlSi10Mg powder addition decreased, which led to a reduction in the gloss of the cladding layer, and the colour became darker.

$$2WC \rightarrow W_2C + C \tag{2}$$

$$3C + 2O_2 \rightarrow 2CO + CO_2 \tag{3}$$

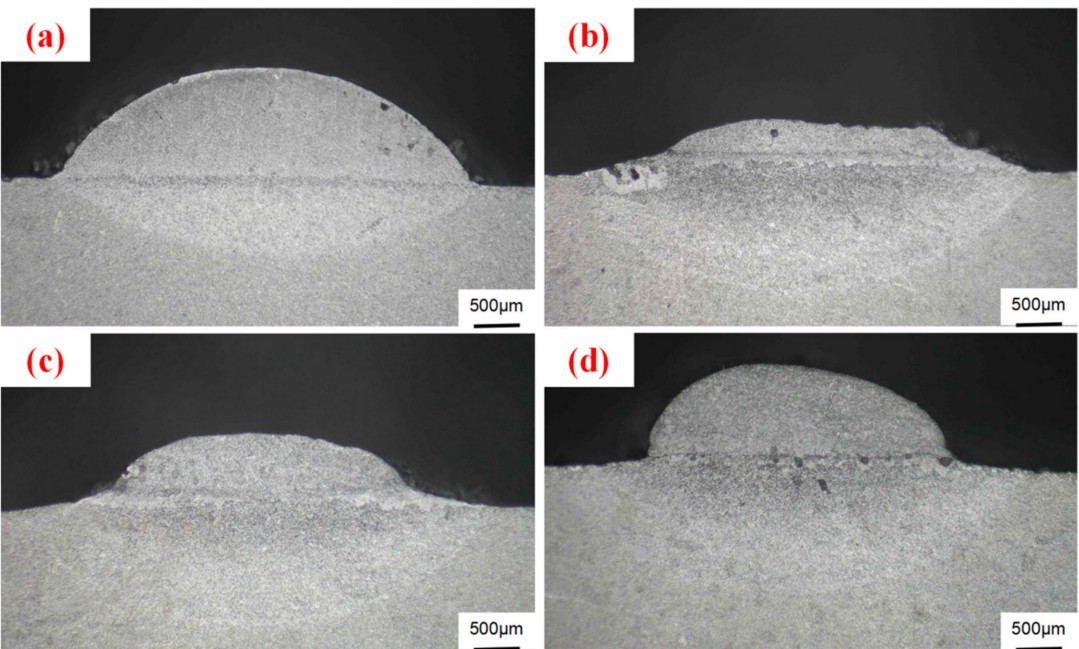

**Figure 4.** Macroscopic morphology of the cross-section of different powder coatings: (**a**) AlSi10Mg coating; (**b**) AlSi10Mg + 20% WC coating; (**c**) AlSi10Mg + 40% WC coating; (**d**) AlSi10Mg + 60% WC coating.

This also contains a small amount of metal gas evaporated at high temperatures, these gases in the molten pool solidification process cannot come up to overflow and remain in the coating to form pores.

The dilution rate is a key parameter to judge the quality of the cladding layer. If the dilution rate is small, it will lead to poor metallurgical bonding between the cladding layer and the substrate, and it is easy to peel off; if the dilution rate is too large, it will lead to a large number of elements in the substrate precipitation into the cladding layer, which will weaken the performance of the cladding layer. The cross-section of the cladding layer is shown in Figure 5, and the dilution rate is calculated as shown in Equation (4).

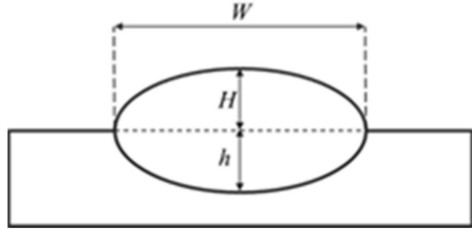

**Figure 5.** Schematic cross-section of the coatings.

Where W is the melt width of the melt pool, H is the height of the cladding layer, h is the height of the melt pool, and μ is the dilution rate of the cladding layer.

$$\mu = \frac{H}{H + h} \times 100\% \qquad (4)$$

Figure 6 shows the parameters such as melt width, melt height, melt depth, and dilution rate for four different powder ratios. It can be seen that, in the case of four different powders coated with the optimal process parameters under orthogonal optimisation, the melting height tends to increase with the increase in WC content, and the melting depth and dilution rate also increase, which is due to the high metallic gloss of aluminium, laser cladding, laser, and aluminium substrate contact with a large number of light sources' refraction occurs in the process of laser coefficient utilisation is poorer, and WC joining makes the reduction of the metallic gloss, the laser coefficient of utilisation rate increases, the temperature in the cladding layer rises, the depth of fusion becomes deeper, and the dilution rate increases.

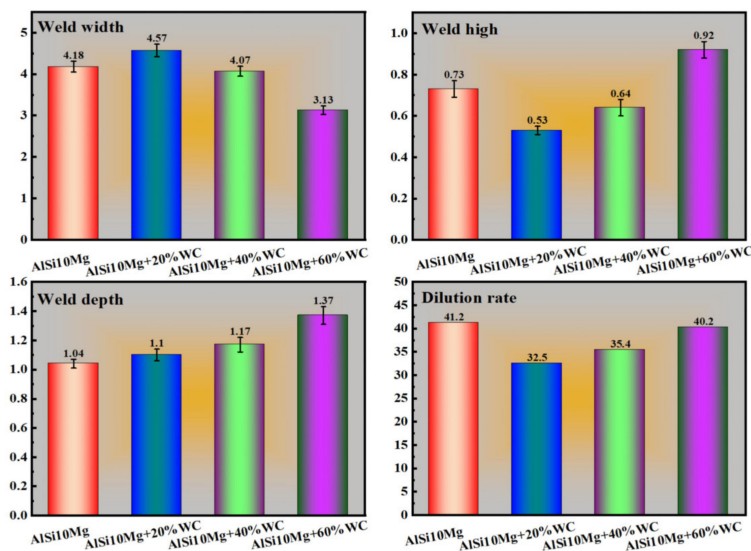

**Figure 6.** Melt width (mm), melt height (mm), melt depth (mm), and dilution rate (%) parameters for four different powder ratios.

*3.2. Phase Analysis*

A German Bruker D8 Advance type X-ray diffraction (XRD) was used to scan the coating for physical analysis with a voltage of 40 kV, a current of 40 mA, a step size of 0.02, and a step rate of 0.2 s/step. the XRD results were then physically analysed using the Jade 6.5 software, and the resultant XRD diffraction pattern is shown in Figure 7.

As can be seen from the figure, there are obvious Al and $Al_2O_3$ diffraction peaks, as well as a small number of SiC diffraction peaks in the coating tissue without the addition of WC ceramic powder; this is due to the fact that a small number of SiC particles in the substrate floats upward in the process of laser fusion cladding, appearing in the substrate and the combination of coatings, Al is very easy to oxidate, generating $Al_2O_3$, the reaction formula is shown in Formula (5). When AlSi10Mg powder is added, the physical phase in the coating changes, and a new diffraction peak appears. WC powder is added, the coating phase changes, new diffraction peaks, and WC firing production of $W_2C$ and C, C and Al and Al's oxide $Al_2O_3$ reaction to generate $Al_4C_3$. The reaction formula is shown in Formula (6), with the WC content continuing increase to 40% WC additions, the coatings appeared in the new phase of $Al_4SiC_4$, which is because with the increase in the WC content of the coatings absorbs more heat, and when the melt pool temperature is higher than 1670 K, SiC in the substrate reacts in situ with Al and $Al_4C_3$ as shown in Formulas (7) and (8), and produces $Al_4SiC_4$ [38]. The increasing W and C

elements decomposed in the melt pool react with the Al and Si atoms to produce more carbides, which have higher hardness, and therefore the microhardness of the coatings is significantly improved compared to the coatings of the no WC addition group, increasing the wear resistance of the coatings. At the same time, the solid solution may produce lattice distortion will also hinder the occurrence of dislocation movement, further enhancing the wear resistance of the cladding layer.

$$2Al + 3O_2 \rightarrow Al_2O_3 \tag{5}$$

$$4Al + 3C \rightarrow Al_4C_3 \tag{6}$$

$$4Al + 4SiC \rightarrow Al_4SiC_4 \tag{7}$$

$$SiC + Al_4C_3 \rightarrow Al_4SiC_4 \tag{8}$$

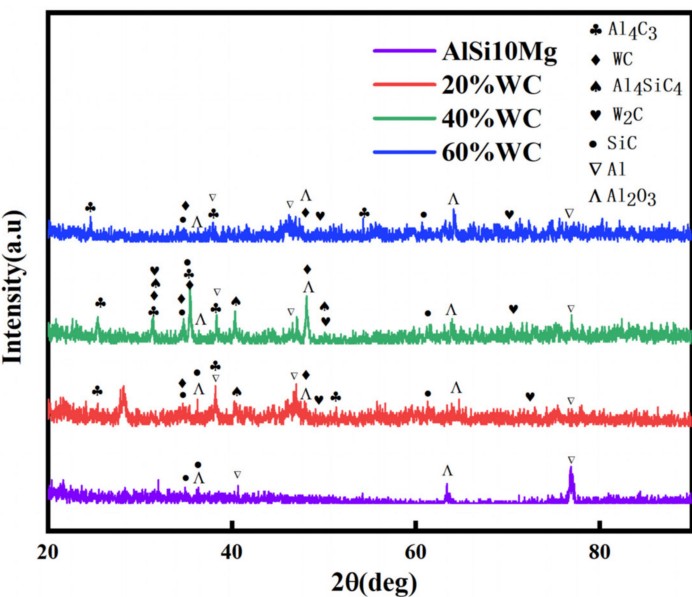

**Figure 7.** XRD patterns of coating with different WC additions.

### 3.3. Microstructure Analysis

JSM-7610F plus Scanning Electron Microscope (SEM) and energy spectrometer (EDS) were used to observe the micro-morphology and elemental distribution. The different areas of the three coatings were scanned for micro-morphology using SEM and Figure 8 shows the micro-morphological microstructure of the four fused coatings.

From Figure 8a–c, it can be seen that the microstructure of the top of the AlSi10Mg cladding layer mainly consists of a large number of granular cellular grains; the microstructure of the middle of the cladding layer shows dendritic grains with an upward growth direction, and the bottom of the cladding layer consists of rod-like cytosine crystals, which contain SiC particles in the substrate, which is mainly due to the fact that the laser cladding is a fast warming and cooling process, and the top of the cladding layer dissipates heat to all sides after laser cladding, and there is no restriction on the direction of grain growth so that granular cytosine is formed. In the solidification process of laser cladding, the microscopic grain structure is related not only to the temperature gradient (G) between the solid–liquid interface but also to the rapid solidification rate (S) during the melting process, where the value of S gradually increases from the bottom to the top of the coating. The larger the G/S value is, the coarser the grain structure is, while the smaller the G/S value, the finer the grain structure [39].

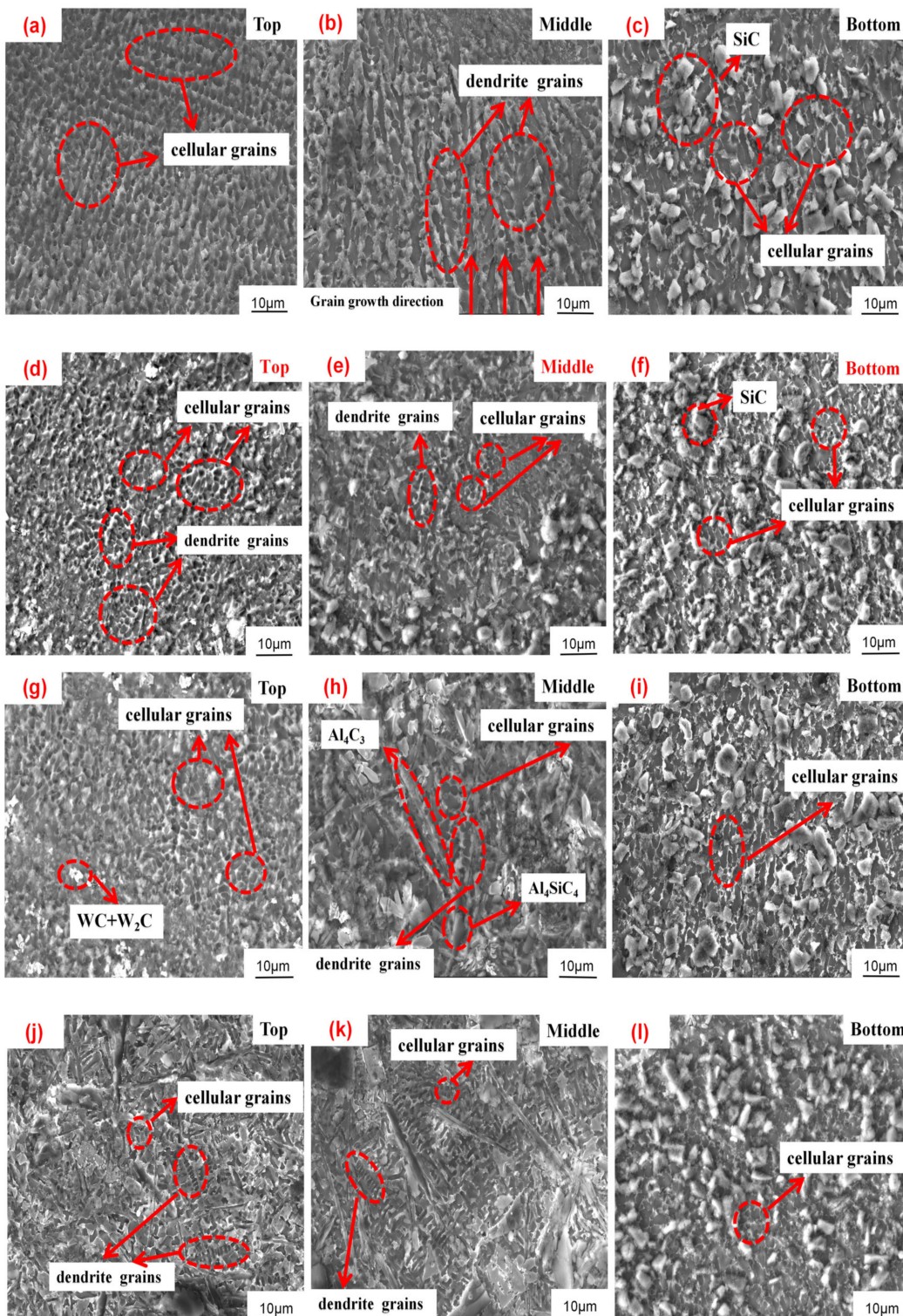

**Figure 8.** Micro-morphological images of four coatings: (**a**–**c**) AlSi10Mg; (**d**–**f**) AlSi10Mg + 20% WC; (**g**–**i**) AlSi10Mg + 40% WC; (**j**–**l**) AlSi10Mg + 60% WC.

As can be seen from Figure 8d–i, when WC was added into AlSi10Mg powder as a reinforcing phase, the morphology of the grains changed due to the change of the physical phase, and the production of the secondary crystal axes started, which was due to the fact that after the addition of WC, a lot of metal compounds and carbides were produced, such as $W_2C$, $Al_4C_3$, and $Al_4SiC_4$, which altered the original microstructure of the AlSi10Mg coating. Comparing Figure 8a,d,g, the morphology of the cellular grain at the top of

the coating changed after the addition of WC, the cellular grain was no longer regular granular but appeared honeycomb, and the carbides such as WC and $W_2C$ appeared in the coating, and the cellular grain appeared in the middle of the layer, which can be seen in Figure 8b,e,h, which indicated that the intermetallic compounds and carbides formed by the addition of WC particles and their decomposition. These compounds provide more efficient and effective solutions to the problems of the AlSi10Mg coatings, and they are not only the most important for the fusion of the AlSi10Mg and $Al_4SiC_4$ layers, but carbide, these compounds provide more nucleation sites, which leads to non-uniform nucleation during solidification, hinders the growth of dendrites and obtains smaller grains, refines the grains of AlSi10Mg coating, generates the needle-like substance $Al_4C_3$ [38], and generates the flaky new phase of $Al_4SiC_4$ when the content of WC reaches 40%; however, the WC also leads to the emergence of a large thermal expansion coefficient, reducing the mobility of the melt pool and generating a small number of porosities. Comparing the bottom Figure 8c,f,i, it can be clearly seen that the grains are refined, on the one hand, due to the addition of WC to refine the grains, and on the other hand, due to the addition of WC to reduce the metal lustre of the molten pool absorbed more heat inside the molten pool, and this heat provides energy for the new nuclei.

As can be seen from Figure 8j–l, When the WC content reaches 60%, significant changes occur in the grain morphology of the cladding layer. A considerable amount of WC forms $W_2C$ compounds, leading to the formation of numerous new nucleation points around $W_2C$ and $Al_4C_3$. In addition to vertically growing cell crystals and dendrites along the growth direction, transverse dendrites also develop at the top of the cladding layer due to newly generated nucleation points that impede vertical grain growth. The presence of solid solutions containing aluminum (such as W, Mg, and Cu atoms) in the cladding layer contributes to solution strengthening, resulting in a shift in grain growth direction within the middle region of the cladding layer and inducing grain dislocation. Consequently, grains grow in multiple directions while interactions between phase interfaces and grains exert a pinning effect, thereby refining and fining down microstructure.

### 3.4. Distribution of Element

The energy spectrum analysis and scanned elemental distribution of the two types of coatings, AlSi10Mg and AlSi10Mg + WC, were carried out by using an energy spectrometer EDS, and the results of energy spectrum analysis and their elemental distributions of the regional surface scans of the AlSi10Mg and AlSi10Mg+WC coatings are shown in Figures 9 and 10, respectively.

From Figure 9 EDS surface scan of AlSi10Mg coating, we can see the distribution of each element, the main distribution of the coatings has Al, Si, C, Mg, and Cu elements, as well as some oxides, in which Mg, Cu elements are diffusely distributed in the whole area, and Mg, Cu elements originally belong to the matrix, which indicates that under laser conduction, the convection in the molten pool makes the elements in the matrix metallurgical bonding with the cladding layer. Therefore, Mg and Cu elements can be detected in the whole cladding layer. Combined with the EDS spectroscopy detection, the results are shown in Table 9, EDS spectroscopy scanning A1 is rich in Si and C elements, and the atomic ratio is close to 1:1, it can be deduced that it is SiC particles, the at.% ratio of Al and O is also close to 2:3.

It can be deduced that it is oxides such as $Al_2O_3$, A2 cellular grains are mainly dominated by the elements of Al and C, and the grain boundaries of A3 are rich in the element of Si, The A3 grain boundary is rich in Si elements. It can be inferred that grains are Al-solid solution, and solid solution particles are Si, C, Cu, Mg, and other elements, due to the high temperature of laser cladding and convection in the molten pool, the SiC particles in the matrix floated upward, and the unmelted SiC particles appeared in the middle of the cladding layer, which improved the hardness and wear resistance of the coating.

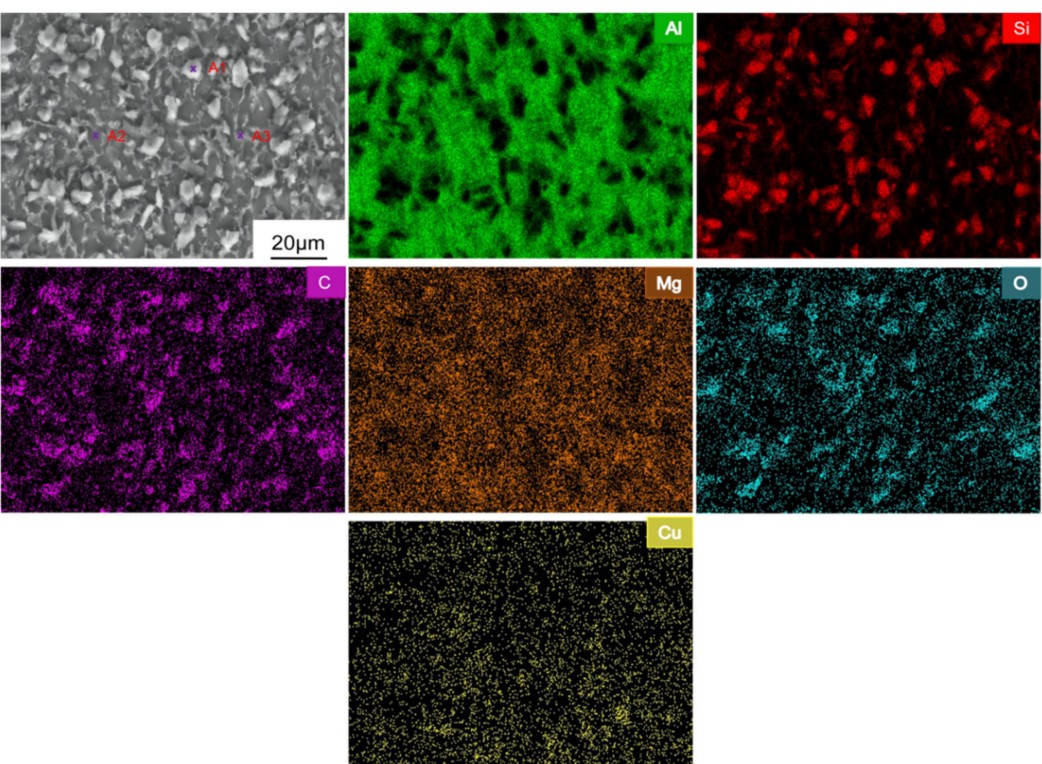

**Figure 9.** EDS map analysis of each element of AlSi10Mg coatings; (**A1**) SiC point scan; (**A2**) Cellular crystal point scan; (**A3**) Grain boundary point scan.

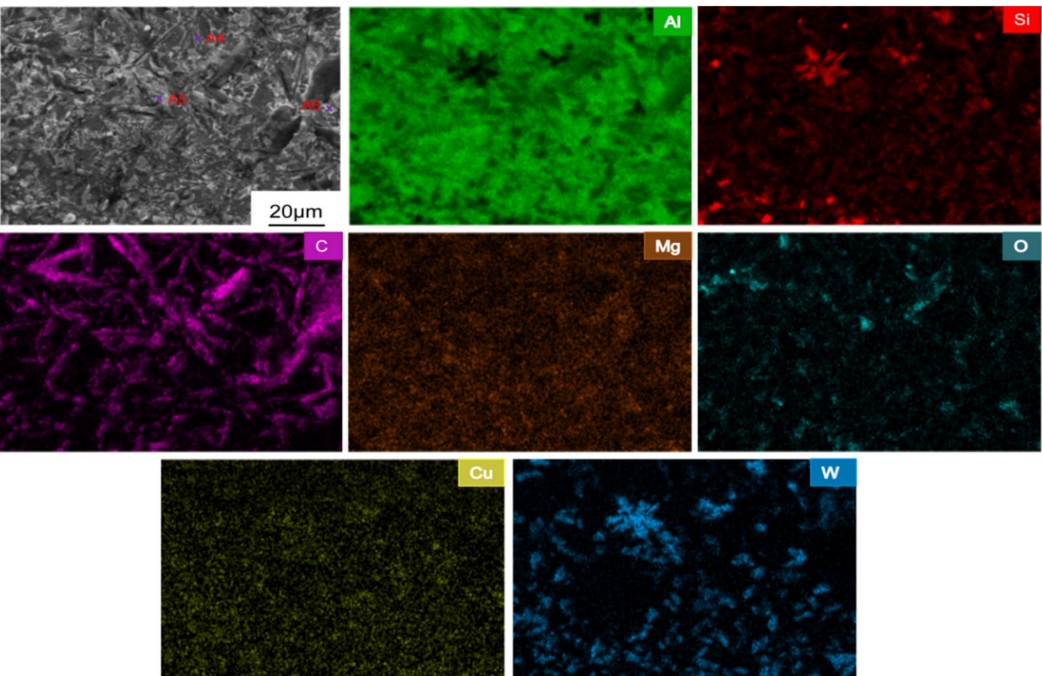

**Figure 10.** EDS map analysis of each element of AlSi10Mg+WC coating: (**A4**) $Al_4C_3$ point scan; (**A5**) Carbide such as WC and $W_2C$ point scan; (**A6**) $Al_4SiC_4$ point scan.

As can be seen from Figure 10 EDS surface scan of AlSi10Mg+WC coatings. After the addition of WC, WC decomposed under the thermal effect of the laser beam and the liquid molten pool, and new substances were generated. It can be seen that the Mg and Cu elements are still diffusely and uniformly distributed in the coatings, the C element is enriched on the needle-like precipitates, and the W element and the Si element are also

distributed in the coatings, but they appear to be enriched in some areas. The results are shown in Table 10 in combination with the EDS energy spectrum.

**Table 9.** Energy spectrum detection of AlSi10Mg coatings.

|  |  | Al | Si | C | O | Cu | Mg |
|---|---|---|---|---|---|---|---|
| A1 | wt.% | 14.44 | 50.25 | 28.75 | 6.56 | - | - |
|  | at.% | 9.34 | 37.25 | 38.32 | 15.09 | - | - |
| A2 | wt.% | 74.87 | 3.68 | 7.90 | 1.32 | 10.58 | 1.64 |
|  | at.% | 71.51 | 3.38 | 16.95 | 2.13 | 4.29 | 1.74 |
| A3 | wt.% | 61.63 | 18.45 | 9.13 | 6.94 | 2.36 | 1.48 |
|  | at.% | 52.49 | 17.09 | 17.89 | 10.22 | 0.87 | 1.44 |

**Table 10.** Energy spectrum detection of AlSi10Mg + WC coatings.

|  |  | Al | Si | C | O | Cu | Mg | W |
|---|---|---|---|---|---|---|---|---|
| A4 | wt.% | 49.18 | 5.51 | 23.49 | 5.46 | 7.37 | 2.35 | 6.64 |
|  | at.% | 43.25 | 3.68 | 37.75 | 6.20 | 2.33 | 1.38 | 5.41 |
| A5 | wt.% | 18.96 | 14.28 | 12.22 | 7.31 | - | - | 47.23 |
|  | at.% | 23.88 | 17.28 | 34.59 | 15.52 | - | - | 8.73 |
| A6 | wt.% | 51.34 | 7.24 | 20.41 | 12.35 | 2.43 | 3.43 | 2.80 |
|  | at.% | 41.25 | 8.68 | 46.75 | 1.13 | 0.40 | 0.38 | 1.41 |

From the EDS spectrum, A4 area elements can be seen, Al elements and C elements are heavily enriched, and the atomic ratio is approximated to 4:3. By combining with the XRD spectrum, we can see that the needle-like material is $Al_4C_3$, A5 area atoms of W content of 8.73 at.%, and C elements of the atomic content of 34.59 at.%;The reason may be that the melting pool temperature exceeds the melting point of WC particles, which makes part of the WC particles dissolve, and some of the W elements are burned, generating unfused WC and carbides such as $W_2C$. The A6 area can be seen through the spectral analysis of the atomic mass ratio of the Al, Si, and C elements, about 4:1:4, which can be roughly judged to be $Al_4SiC_4$, the morphology is a lamellar structure, and the hardness and abrasion resistance of coatings are both significantly improved by $Al_4C_3$ and $Al_4SiC_4$.

*3.5. Microhardness Analysis*

Huayin HV-1000A digital microhardness tester was used to measure the microhardness of the substrate and the three coatings, with an applied force of 200 N and a dwell time of 15 s. Figure 11 shows the Vickers microhardness distribution of the coatings of AlSi10Mg powder and different proportions of the WC ceramic powder additions in a line graph, and the microhardness values were measured from the top of the coatings to the direction of the substrate depth at intervals of 0.2 mm. The average hardness value of the three points is taken as the final hardness value of the point, and two more points are taken above and below the same point for three measurements. The hardness map can be divided into two parts from left to right: the cladding zone (CZ) and the substrate (SZ). Overall, the distribution of hardness curves for different WC additions to the coatings has a similar trend of increasing and then decreasing, while the AlSi10Mg powder hardness line graph is gradually increasing. The hardness of the added WC ceramic powder is greater in the coatings zone, followed by the heat-affected zone, and the substrate is the lowest, while the AlSi10Mg coating is the coatings zone where the hardness is lower than the substrate hardness.

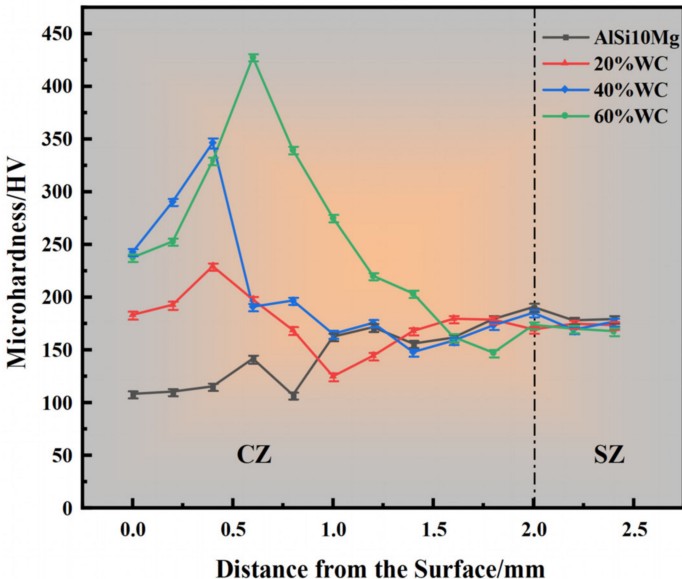

**Figure 11.** Microhardness distribution curve.

It can be seen from the figure that the highest hardness point did not occur in the surface layer. This is because, in the cladding process, the surface layer of the powder has been absorbing laser energy, generating a large amount of heat, in which the alloying elements Al, W, etc., at high temperatures sintering occurs, and the surface hardness is reduced. The hardness of the upper part of the cladding layer is higher than the lower part because the upper part of the cladding layer is in contact with the atmosphere, the cooling crystallisation rate is fast, the grain size is small, and the formation of fine crystalline reinforcement and increase hardness; add WC ceramic powder, the cladding layer microhardness are higher than the substrate, this is because of the addition of the WC powder and the substrate in the high-temperature melting of the SiC. The WC decomposition out of the W atoms and C atoms solidly dissolved in the formation of the Al substrate solid solution.

Strengthening enhances the hardness of the cladding layer, the decomposition of C, Si to form $Al_4C_3$, $Al_4SiC_4$, and other hard phases, the formation of the second strengthening, uniformly distributed in the cladding layer, and the formation of diffuse strengthening to further enhance the cladding layer microhardness. The hardness of the cladding layer in the figure has a large gap in which the average hardness of the matrix, AlSi10Mg powder, 20% WC, 40% WC, and 60% WC powder cladding layer is 171.61 HV, 124.20 HV, 182.74 HV, 283.77 HV, and 310.37 HV, respectively, compared to the hardness of the matrix of 171.61 HV, 20% WC cladding layer. The hardness of 20% WC coatings increased 1.06 times, 40% WC coatings increased 1.65 times, and 40% WC coatings increased 1.8 times, while the hardness of AlSi10Mg powder coatings was lower than that of the substrate, which is because the substrate contains SiC hard particles. The coefficient of thermal expansion of SiC particles is lower than the coefficient of thermal expansion of aluminium, and in the solidification process, the accumulation of SiC particles around the particles will generate a large number of thermal. During solidification, the SiC particles will accumulate around the SiC particles to produce a large number of heat-altering dislocations, and at the same time, the SiC particles will inhibit the grain growth of aluminium crystals and reduce the grain size of the aluminium alloy.

### 3.6. Wear Performance Analysis

The friction factor of the coating was tested by MFT-5000 friction and wear tester, the friction and wear curves obtained are shown in Figure 12.

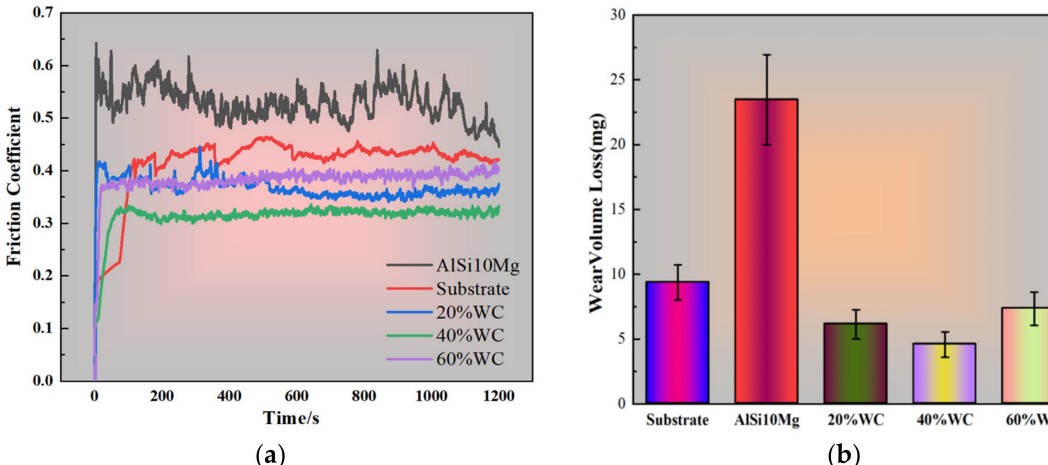

**Figure 12.** (**a**) Friction wear curves for different coatings; (**b**) wear weight loss diagram for different coatings.

Figure 12a,b show the friction and wear curves of several different coatings and substrates over time and the wear loss graphs before and after the specimen tests, respectively. It can be seen that there are two stages of initial run-in and steady wear for several coatings. Among them, the friction wear curve of AlSi10Mg coating has been in a large fluctuation state, and the wear curve of the substrate tends to stabilise after about 400 s, the wear curve of the coating with 20% WC added tends to stabilise after about 500 s, the wear curve of 40% WC tends to stabilise around 200 s, and the wear curve of 60% WC goes to the steady stage around 20 s. This is due to the fact that AlSi10Mg coating hardness is low, from the initial stage of frictional wear coating which has a large amount of wear, $Si_3N_4$ ceramic ball friction vice, and the sample surface of the micro-convex body of the collision between friction, the contact stress caused by the friction coefficient fluctuations. The convex body was worn flat, the ceramic ball and the coating of the contact area increased, the pit deepened, the friction between the contact surface also increased, and the coefficient of friction is the largest of the groups of data and the least resistant.

The largest, but also the least wear-resistant group due to the substrate is 15% SiCp/2009Al. The substrate contains SiC ceramic particles, the hardness is enhanced, and the surface roughness increases; the friction factor cannot tend to a stable stage in a short time, and occasionally there are fluctuations, it may be the friction side of the ceramic ball in contact with the SiC ceramic particles, the coating will play a role in the addition of 20% WC-40% WC solid solution strengthening and diffusion strengthening role, the generation of hard phase carbide improves the hardness and wear resistance of the coating and the friction factor is also reduced.With the increase in WC content, the hardness continues to improve and the coating surface will be a flaking phenomenon, increasing the friction between the friction sub and the coating, but instead of the friction factor appearing in the 60% WC additions to the coating is higher than the 40% WC content of the coating. In turn, it can be shown that the hardness is not the higher the better.

From the tests, it can be seen that under the same load, the average coefficient of friction of 15% SiCp/2009Al substrate, AlSi10Mg coating, AlSi10Mg + 20% WC coating, AlSi10Mg + 40% WC coating, and AlSi10Mg + 60% WC coating are 0.54, 0.43, 0.39, 0.31, and 0.36 specimens, respectively. The observation was consistent with the friction coefficient results that specimens with a lower friction coefficient also exhibited a lower wear loss [40]. The wear was 9.36 mg, 23.47 mg, 6.13 mg (34.5% reduction compared to the substrate), 4.58 mg (51.06% reduction compared to the substrate), and 7.35 mg (21.47% reduction compared to the substrate), respectively.

The SEM microscopic morphology of 15% SiCp/2009Al substrate, AlSi10Mg coating, AlSi10Mg + 20%WC coating, AlSi10Mg + 40%WC coating, and AlSi10Mg + 60%WC coating at different magnifications after surface friction are shown in Figure 13a–j, respectively, and

the main wear in the coatings is the abrasive wear and the adherent wear, as can be seen in Figure 13a,b.The wear characteristics of the substrate are mainly flaking and friction grooves there are some irregular pits and a small number of plough grooves with attached abrasive debris on the abrasive marks on the surface of the substrate. This is because the matrix and the friction pair contact adhesion and plastic deformation.

In Figure 13c,d, it can be seen that the surface of the coating has a large number of flaking and plough grooves, and there is a slight depression, the coating is very soft in texture, and the abrasion resistance is Very poor, $Si_3N_4$ ceramic ball friction vice adhesion with the coating, adhesion point tearing, plastic deformation of the coating; from Figure 13e,f, it can be clearly seen that the coating grooves become shallower, accompanied by a small amount of spalling; this is because of the addition of WC in the coating to make the hardness of the coating increased, compared with the substrate and AlSi10Mg coating wear resistance increased.

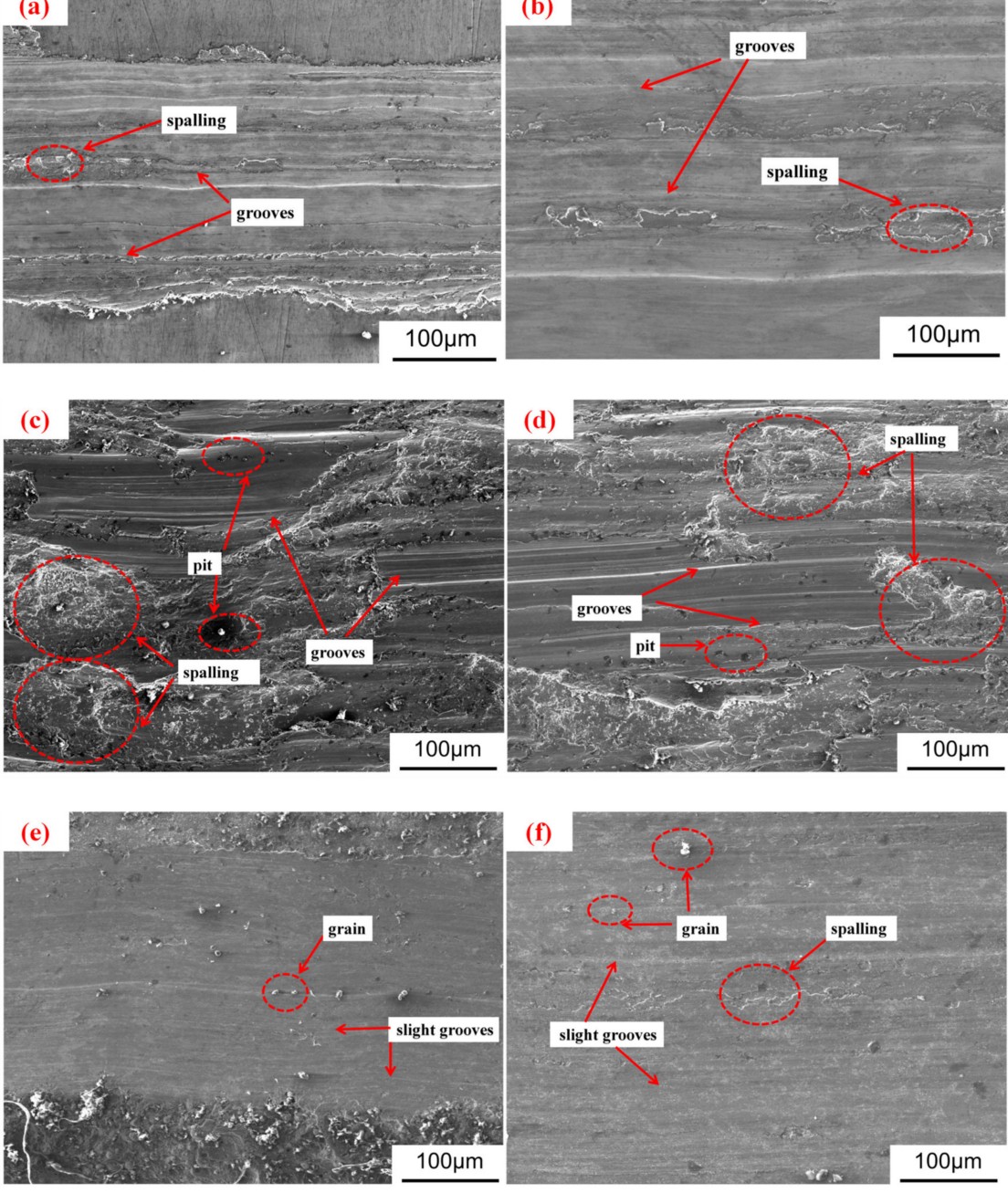

**Figure 13.** *Cont.*

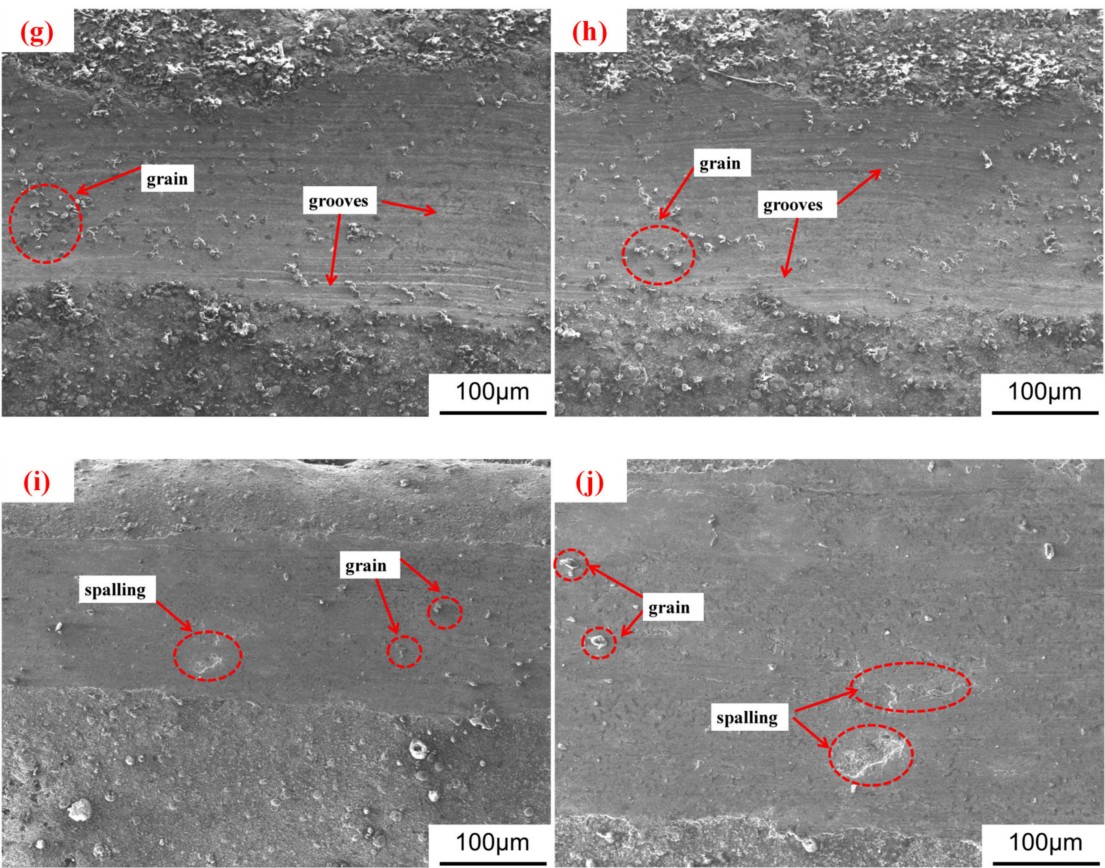

**Figure 13.** Wear morphology and microstructure of four coatings: (**a**,**b**) substrate; (**c**,**d**) AlSi10Mg; (**e**,**f**) AlSi10Mg + 20% WC; (**g**,**h**) AlSi10Mg + 40% WC; (**i**,**j**) AlSi10Mg + 60% WC.

Figure 13g,h show that the content of WC continues to increase and the hardness continues to increase; due to the high hardness, there is no spalling, only slight scratches and some particles, which is due to the production of hard compounds in the coating, so that the wear mechanism becomes abrasive wear and the abrasive marks become narrower, with better abrasion resistance. Figure 13i,j show in the 60% WC additions, although it makes the coating hardness higher and there are no obvious grooves, there is a flaking and the coating toughness decreases, increasing the possibility of coating cracking, and the performance is rather reduced [40].

Roughness is also an important parameter to reflect the quality of coating fusion. The higher the roughness [41], the softer the coating texture, the increased contact area between the friction vice and the coating surface, the wear resistance is reduced, and the quality of the coating decreases. The 3D contour of the friction wear was obtained by using the super depth of field microscope, and the ends of the friction wear grooves of the coated coating were marked with two points and analysed, respectively, for 15% SiCp/2009Al substrate, AlSi10Mg coating, AlSi10Mg + 20% WC coating, AlSi10Mg + 40% WC coating, and AlSi10Mg + 60% WC coating. WC coating, AlSi10Mg + 40%WC coating, and AlSi10Mg + 60%WC coating. The analysed morphology diagram is shown in the Figure 14a–e.

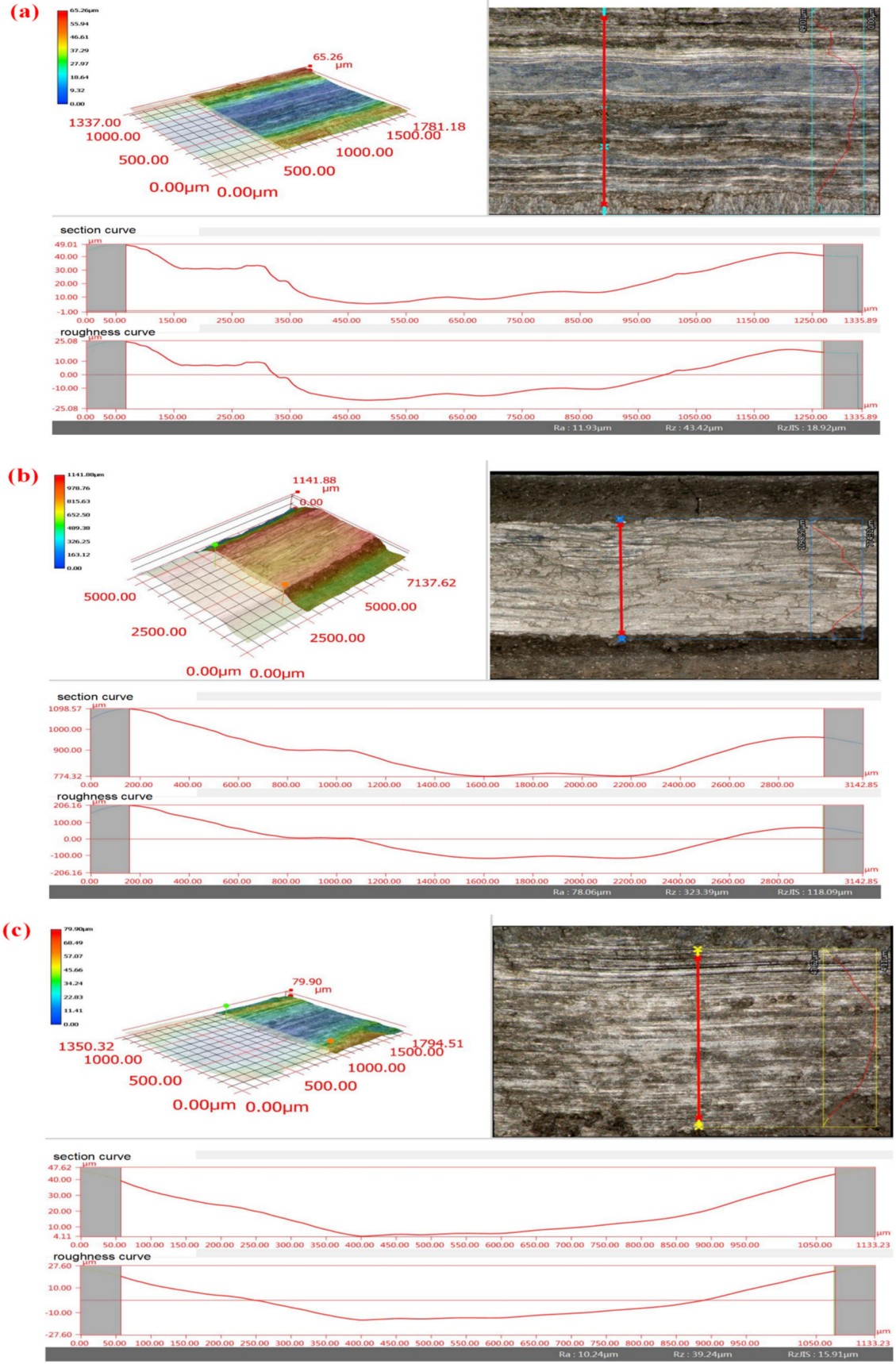

**Figure 14.** *Cont.*

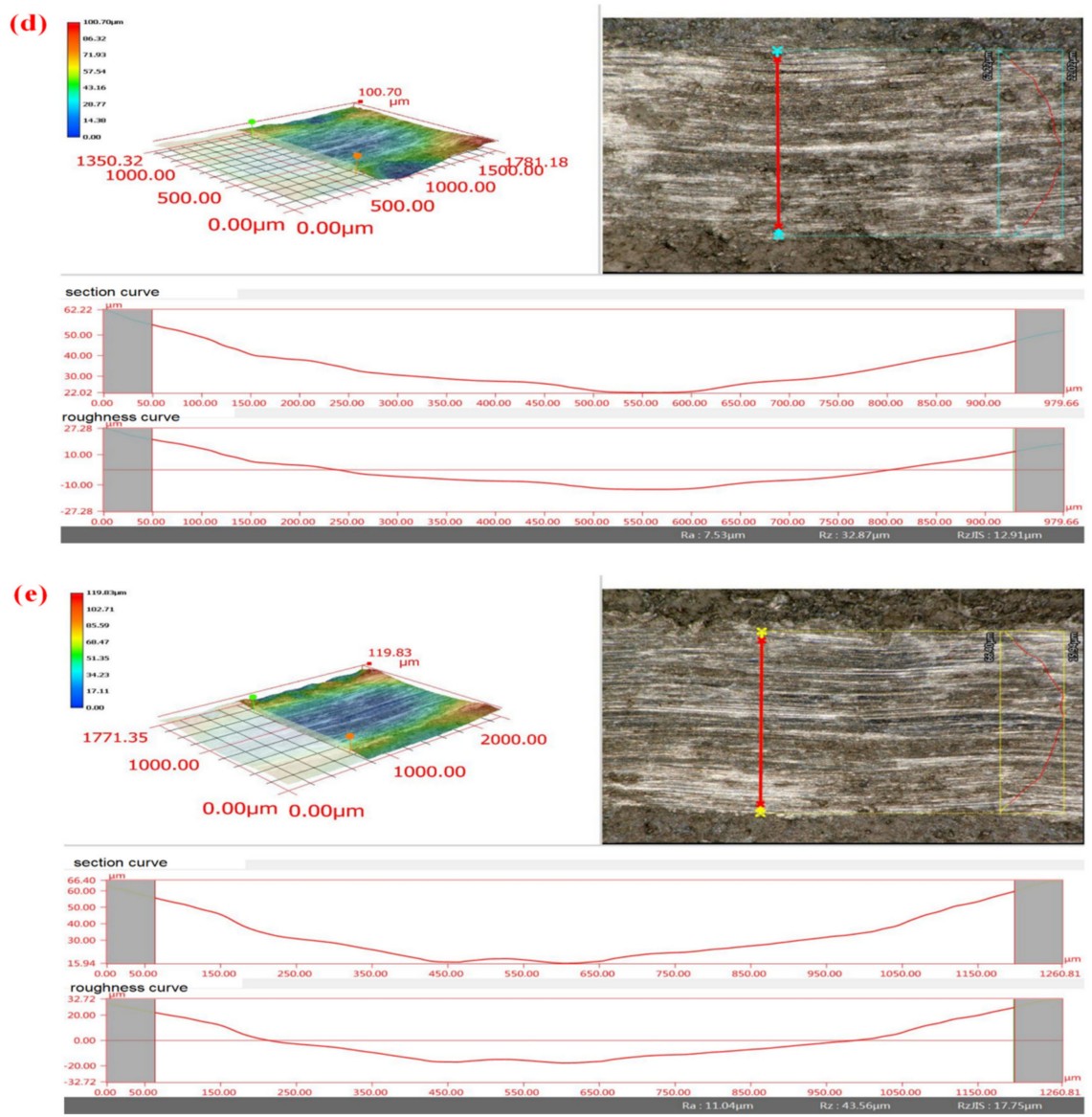

**Figure 14.** Friction and wear morphology and microstructure of the four coatings. (**a**) Substrate; (**b**) AlSi10Mg; (**c**) AlSi10Mg + 20%WC; (**d**) AlSi10Mg + 40%WC; (**e**) AlSi10Mg + 60%WC.

From the above Figure 14, it can be seen that the coating cross-section curves are all low in the middle and high on both sides, showing a concave tendency; the concave tendency is $Rz(b) > Rz(e) > Rz(a) > Rz(c) > Rz(d)$ in descending order, the highest degree of concavity is 323.39 μm for the AlSi10Mg coating $Rz(b)$, and the lowest degree of concavity is 32.87 μm for the AlSi10Mg + 40% WC coating $Rz(d)$ 32.87 μm; and the roughness size from high to low is $Ra(b) > Rz(a) > Ra(e) > Ra(c) > Ra(d)$, the largest coating roughness is 78.06 μm for AlSi10Mg coating $Ra(b)$, and the smallest one is 7.53 μm for AlSi10Mg + 40%WC coating $Ra(d)$. The analysis of the two data sets and the coating weight loss Figure 14b reveals that the AlSi10Mg coating exhibits low hardness, with a significant accumulation of coating powder in the friction pair resulting in high weight loss. Consequently, the coating demonstrates poor hardness and limited wear resistance. However, upon incorporating 40% WC ceramic powder into the coating, there is a substantial increase in its hardness while minimising powder adhesion in the friction pair and reducing weight loss to a minimum. This optimised performance can be attributed to WC powder's higher melting point compared to AlSi10Mg powder coating.

## 4. Conclusions

(1) The exploration of laser cladding process parameters and the optimisation of orthogonal tests through the selection of the lowest dilution rate and suitable melt width, as well as a comprehensive analysis of the macro-morphology, the optimal process parameters were finally determined to be $P$ = 4400 W, $V_f$ = 11.3 g·min$^{-1}$, and $V_S$ = 1800 mm·min$^{-1}$.

(2) By incorporating WC enhanced phase, the cladding layer facilitates the reaction between metal elements and WC, resulting in an abundance of hard compounds within the coating. This leads to the emergence of new phases such as $W_2C$, $Al_4C_3$, and $Al_4SiC_4$ in the cladding layer. Additionally, there is an increased presence of nucleation particles and a distribution of WC at grain boundaries. Consequently, this delays grain growth and refines the microstructure of the coating, ultimately improving its overall quality. Simultaneously, by introducing solid solution strengthening through aluminum's incorporation with atoms like W, Mg, and Cu in the cladding layer; it causes a shift in grain growth direction within the middle region of the said layer while inducing dislocation among grains. The interaction between phase interfaces and grains exhibits a pinning effect that further enhances microstructural refinement by promoting multi-directional grain growth.

(3) The average hardness and abrasion resistance of the coatings increased with the increase in WC additive amount, and when the additive amount was 40%, the macroscopic morphology was good, and no spalling was found in the friction and wear morphology, and the lowest friction factor, which indicated that the abrasion resistance was the best; however, when the additive amount was 60%, there were obvious macroscopic cracks in the coatings and obvious spalling in the friction and wear morphology, and the quality of the coatings was reduced. The result is consistent with experimental predictions.

**Author Contributions:** Conceptualization, X.F.; formal analysis, X.F. and X.L.; methodology, X.F.; investigation, X.F. and F.W.; data curation, X.F. and Z.L.; writing—original draft, X.F.; writing—review and editing, X.F. and F.W; visualization, Z.L. and W.W. All authors have read and agreed to the published version of the manuscript.

**Funding:** This research was funded by the Tianchi Doctor of Philosophy Program of Xinjiang Uygur Autonomous Regions (No. 610221023).

**Institutional Review Board Statement:** Not applicable.

**Informed Consent Statement:** Informed consent was obtained from all subjects involved in this study.

**Data Availability Statement:** Data are contained within the article.

**Conflicts of Interest:** The authors declare no conflict of interest.

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
