# Peer review of "Effect of Spherical WC Content on the Microstructure and Properties of SiCp Aluminium Composite Material"

_coatings, doi:10.3390/coatings13111935_

Round 1

Reviewer 1 Report

Comments and Suggestions for Authors

The paper can be accepted, but it is necessary to carry out certain refinements in order to improve the quality of the work and its application for subsequent tests.

-          Expand the introduction or other chapters with papers from this field.

-          It is necessary to review and cite recent works from this and similar fields, such as:

https://doi.org/10.3390/coatings13081379

http://dx.doi.org/10.24874/PES01.01.029

https://doi.org/10.18485/aeletters.2019.4.3.1

-          At the end of the introduction, state what is the main contribution of the work and how does this work differ from similar works in this field? What is the reason, ie. why should this work be published? 

-          Increase the number of references in the work based on the given remarks.

-          In chapter 3.1. it is necessary to correctly label certain oxides and mixtures with chemical formulas, such as Al3O2 and Al4SiC4, instead of Al3O2 and Al4SiC4, as stated in the text.

-          In chapter 3.5, it is necessary to describe in more detail the possibilities and characteristics of the Huayin HV-1000A digital microhardness tester, with an indication of its working range and why this tester was chosen for measurement, and if possible, to provide its photo.

-          In chapter 3.6, it is necessary to describe in more detail the capabilities and characteristics of the MFT-5000 friction and wear tester, stating its working range and why this tester was chosen for measurement, and if possible, provide its photo.

-          It is necessary to state whether the obtained experimental results can be described by some of the mathematical models (Taguchi, TOPSIS, PROMETHEE and similar methods), i.e. whether it is possible to predict the obtaining of certain sizes and parameters and other experimental conditions before and after applying the layer with some mathematical model with a laser on an alloy with a different percentage of additives in a certain range, and not only for precisely defined values ​​of the share of additives of 40% and 60% (for example, coefficients of friction, degree of surface wear, hardness at the micro level, temperature changes when using different types and percentages of additives and layers that are applied to the basic aluminum alloy as well as other physical sizes, with reference to earlier works).

-          In the concluding remarks, it is necessary to unequivocally state whether the obtained results confirmed the expectations and assumptions, i.e. whether there were deviations or completely opposite results from the expected ones.

-          Based on the analysis and discussion, expand the concluding considerations, especially in terms of further research in the field of application of other materials and additives, both for the basic aluminum alloy and for the layer that is applied to improve the characteristics of the basic alloy, but also in the field of application of other modern the technique of applying surface layers of materials, using lasers of different characteristics or other production and diagnostic equipment. In accordance with the current possibilities, it is necessary to comment on the expected costs and the price of the application of this technological procedure, primarily because of the possibility of its commercialization and mass application, with indications of future research in this field.

Reviewer 2 Report

Comments and Suggestions for Authors

After going through the paper titled "Effect of Spherical WC Content on the Microstructure and Properties of SiCp Aluminium composite material"

1. There is lack of novelty in the abstract. Quantified data is absent.

2. Lines 82-89 cannot be understood at all. Please revise the entire paragraph concluding the literature review, the research gap and the methodology to be followed in brief.

3. The English needs correction. For example, lines 101-104 is unnecessary long. There is too much use of 'and' due to which it becomes difficult to follow what the authors are trying to say.

4. The basis of the parameters shown in Table 4 is not clear. What is meant by "The process parameters were optimised by orthogonal tests, and the  values are shown in Table 4."

5. Lines 131-134 is unclear. What is an orthogonal array? It seems to be Taguchi's L9 array. Why was it chosen? 

6. The parameters laid down in Table 5 does not have interaction effects? The interaction effects cannot be accommodated in the L9 array. SO why authors have chosen this?

7. What is the basis of the tribo-test parameters in lines 201-204.

8. The Figure 7 is very unclear. Please redraw in white background. Moreover, the lines 269-272 should be in experimental details.

9. Lines 322-323 is incomplete. Also lines 355-357 is incomplete. This is the case at several places in the manuscript.

10. Lines 379-384, the authors state that wear resistance improved. But the wear results have not been discussed previously. Is it being compared with some other paper?

11. Lines 456-560, shouldn't it be stated in experimental details?

12. After this it becomes very difficult to read the explanations and correlate. Nevertheless, the results seem to be good. But the authors have failed to communicate. Further, there is no comparison with literature. 

Comments on the Quality of English Language

The English is difficult to understand. There are unnecessary length lines which must be shortened. This makes the paper very difficult to understand.

Reviewer 3 Report

Comments and Suggestions for Authors

In this work, authors have performed laser cladding on SiCp aluminum matrix composites using different mixtures of AlSi10Mg and WC powders. They have performed friction and wear tests to show that AlSi10Mg+40%WC coating is in optimal condition. The manuscript is well-written and can be accepted with minor revision. The reviewer has the following comments.

1.       Laser parameters must be provided, such as wavelength, intensity distribution, continuous wave or pulsed laser, etc.

2.       Table 9 can be plotted into a graph for an easier interpretation of how the weld width, height, depth, and dilution rate change with increasing WC content.

3.       In Figure 7, AlSi10Mg and 60% WC have the same color, so it is difficult for the readers to understand. It is recommended to change the color of one line.

4.       How many samples were tested for hardness for each condition? Variations of hardness should be provided with error bars in Figure 11.

5.       How many samples were tested in each category for friction coefficient? Is there a variation of friction coefficient among samples in the same category?

6.       The conclusion is too long and can be shortened.

Round 2

Reviewer 2 Report

Comments and Suggestions for Authors

Most of the technical comments have been taken care of. Though I still feel the sentence structures can be improved. The following modification are still required:

1. The text inside Fig. 11 and Fig. 16 cannot be read. It should be improved. May be a black writing inside white background should serve the purpose.

2. Fig. 5 and Fig. 6 are unnecessary.

3. Fig. 1 and 2 can be combined to a single figure.

Comments on the Quality of English Language

The sentences are unnecessarily long and difficult to follow at times.
